# Adaptation in protein fitness landscapes is facilitated by indirect paths

**Nicholas C Wu[1,2†‡], Lei Dai[1,3†], C Anders Olson[1], James O Lloyd-Smith[3], Ren Sun[1,2*]**

[1]Department of Molecular and Medical Pharmacology, University of California, Los Angeles, Los Angeles, United States; [2]Molecular Biology Institute, University of California, Los Angeles, Los Angeles, United States; [3]Department of Ecology and Evolutionary Biology, University of California, Los Angeles, Los Angeles, United States

*For correspondence: rsun@mednet.ucla.edu

†These authors contributed equally to this work

Present address: ‡Department of Integrative Structural and Computational Biology, The Scripps Research Institute, La Jolla, United States

Competing interests: The authors declare that no competing interests exist.

**Abstract** The structure of fitness landscapes is critical for understanding adaptive protein evolution. Previous empirical studies on fitness landscapes were confined to either the neighborhood around the wild type sequence, involving mostly single and double mutants, or a combinatorially complete subgraph involving only two amino acids at each site. In reality, the dimensionality of protein sequence space is higher ($20^L$) and there may be higher-order interactions among more than two sites. Here we experimentally characterized the fitness landscape of four sites in protein GB1, containing $20^4 = 160,000$ variants. We found that while reciprocal sign epistasis blocked many direct paths of adaptation, such evolutionary traps could be circumvented by indirect paths through genotype space involving gain and subsequent loss of mutations. These indirect paths alleviate the constraint on adaptive protein evolution, suggesting that the heretofore neglected dimensions of sequence space may change our views on how proteins evolve.

## Introduction

The fitness landscape is a fundamental concept in evolutionary biology (*Kauffman and Levin, 1987*; *Poelwijk et al., 2007*; *Romero and Arnold, 2009*; *Hartl, 2014*; *Kondrashov and Kondrashov, 2015*; *de Visser and Krug, 2014*). Large-scale datasets combined with quantitative analysis have successfully unraveled important features of empirical fitness landscapes (*Kouyos et al., 2012*; *Barton et al., 2015*; *Szendro et al., 2013*). Nevertheless, there is a huge gap between the limited throughput of fitness measurements (usually on the order of $10^2$ variants) and the vast size of sequence space. Recently, the bottleneck in experimental throughput has been improved substantially by coupling saturation mutagenesis with deep sequencing (*Fowler et al., 2010*; *Hietpas et al., 2011*; *Jacquier et al., 2013*; *Wu et al., 2014*; *Thyagarajan and Bloom, 2014*; *Qi et al., 2014*; *Stiffler et al., 2015*), which opens up unprecedented opportunities to understand the structure of high-dimensional fitness landscapes (*Jiménez et al., 2013*; *Pitt and Ferré-D'Amaré, 2010*; *Payne and Wagner, 2014*).

Previous empirical studies on combinatorially complete fitness landscapes have been limited to subgraphs of the sequence space consisting of only two amino acids at each site ($2^L$ genotypes) (*Weinreich et al., 2006*; *Lunzer et al., 2005*; *O'Maille et al., 2008*; *Lozovsky et al., 2009*; *Franke et al., 2011*; *Tan et al., 2011*). Most studies of adaptive walks in these diallelic sequence spaces focused on "direct paths" where each mutational step reduces the Hamming distance from the starting point to the destination. However, it has also been shown that mutational reversions can occur during adaptive walks in diallelic sequence spaces such that adaptation proceeds via "indirect paths" (*DePristo et al., 2007*; *Berestycki et al., 2014*; *Martinsson, 2015*; *Li, 2015*; *Palmer et al.,*

**eLife digest** Proteins can evolve over time by changing their component parts, which are called amino acids. These changes usually happen one at a time and natural selection tends to preserve those changes that make the protein more efficient at its specific tasks, while discarding those that impair the protein's activity. However the effect of each change depends on the protein as a whole, and so two changes that separately make the protein worse can make it much better if they occur together. This phenomenon is called epistasis and in some cases it can trap proteins in a sub-optimal form and prevent them from improving further.

Proteins are made from twenty different kinds of amino acid, and there are millions of different combinations of amino acids that could, in theory, make a protein of a given length. Studying protein evolution involves making variants of the same protein, each with just a few changes, and comparing how efficient, or "fit", they are. Previous studies only measured the fitness of a few variants and showed that epistasis could block protein evolution by requiring the protein to lose some fitness before it could improve further. However, new techniques have now made it easier to study protein evolution by testing many more protein variants.

Wu, Dai et al. focused on four amino acids in part of a protein called GB1 and tested the efficiency of every possible combination of these four amino acids, a total of 160,000 ($20^4$) variants. Contrary to expectations, the results suggested that the protein could evolve quickly to maximise fitness despite there being epistasis between the four amino acids. Overcoming epistasis typically involved making a change to one amino acid that paved the way for further changes while avoiding the need to lose fitness. The original change could then be reversed once the epistasis was overcome. The complexity of this solution means it can only be seen by studying a large number of protein variants that represent many alternative sequences of protein changes.

Wu, Dai et al. conclude that proteins are able to achieve a higher level of fitness through evolution by exploring a large number of changes. There are many possible changes for each protein and it is this variety that, despite epistasis, allows proteins to become naturally optimised for the tasks that they perform. While the full complexity of protein evolution cannot be explored at the moment, as technology advances it will become possible to study more protein variants. Such advances would therefore hopefully allow researchers to discover even more about the natural mechanisms of protein evolution.

*2015*). In sequence space with higher dimensionality ($20^L$, for a protein sequence with *L* amino acid residues), the extra dimensions may further provide additional routes for adaptation (*Gavrilets, 1997*; *Cariani, 2002*). Although the existence of indirect paths has been implied in different contexts, it has not been studied systematically and its influence on protein adaptation remains unclear. Another underappreciated property of fitness landscapes is the influence of higher-order interactions. Empirical evidence suggests that pairwise epistasis is prevalent in fitness landscapes (*Kvitek and Sherlock, 2011*; *Kouyos et al., 2012*; *O'Maille et al., 2008*; *Lozovsky et al., 2009*). Specifically, sign epistasis between two loci is known to constrain adaptation by limiting the number of selectively accessible paths (*Weinreich et al., 2006*). Higher-order epistasis (i.e. interactions among more than two loci) has received much less attention and its role in adaptation is yet to be elucidated (*Weinreich et al., 2013*; *Palmer et al., 2015*).

## Results

### Empirical determination of a four-site fitness landscape

In this study, we investigated the fitness landscape of all variants ($20^4$ = 160,000) at four amino acid sites (V39, D40, G41 and V54) in an epistatic region of protein G domain B1 (GB1, 56 amino acids in total) (*Figure 1—figure supplement 1*), an immunoglobulin-binding protein expressed in Streptococcal bacteria (*Sjöbring et al., 1991*; *Sauer-Eriksson et al., 1995*). The four chosen sites contain 12 of the top 20 positively epistatic interactions among all pairwise interactions in protein GB1, as we previously characterized (*Olson et al., 2014*) (*Figure 1—figure supplement 2*). Thus the sequence

space is expected to cover highly beneficial variants, which presents an ideal scenario for studying adaptive evolution. Moreover, this empirical fitness landscape is expected to provide us insights on how high dimensionality and epistasis would influence evolutionary accessibility. Briefly, a mutant library containing all amino acid combinations at these four sites was generated by codon randomization. The "fitness" of protein GB1 variants, as determined by both stability (i.e. the fraction of folded proteins) and function (i.e. binding affinity to IgG-Fc), was measured in a high-throughput manner by coupling mRNA display with Illumina sequencing (see Materials and methods, *Figure 1— figure supplement 3*) (*Roberts and Szostak, 1997*; *Olson et al., 2012*). The relative frequency of mutant sequences before and after selection allowed us to compute the fitness of each variant relative to the wild type protein (WT). While most mutants had a lower fitness compared to WT (fitness < 1), 2.4% of mutants were beneficial (fitness > 1). (*Figure 1—figure supplement 4*). We note that this study does not aim to extrapolate protein fitness to organismal fitness. Although there are examples showing that protein fitness in vitro correlates with organismal fitness in vivo (*Natarajan et al., 2013*; *Wu et al., 2012*), this relation may not be linear and is likely to be system-specific due to the difference in selection pressures in vitro and in vivo (*Pál et al., 2006*; *Hingorani and Gierasch, 2014*).

## Direct paths of adaptation are constrained by pairwise epistasis

To understand the impact of epistasis on protein adaptation, we first analyzed subgraphs of sequence space including only two amino acids at each site (*Figure 1A*). Each subgraph represented a classical adaptive landscape connecting WT to a beneficial quadruple mutant, analogous to previously studied protein fitness landscapes (*Weinreich et al., 2006*; *Szendro et al., 2013*). Each variant is denoted by the single letter code of amino acids across sites 39, 40, 41 and 54 (for example, WT sequence is VDGV). Each subgraph is combinatorially complete with $2^4 = 16$ variants, including WT, the quadruple mutant, and all intermediate variants. We identified a total of 29 subgraphs in which the quadruple mutant was the only fitness peak. By focusing on these subgraphs, we essentially limited the analysis to direct paths of adaptation, where each step would reduce the Hamming distance from the starting point (WT) to the destination (quadruple mutant). Out of 24 possible direct paths, the number of selectively accessible paths (i.e. with monotonically increasing fitness) varied from 12 to 1 among the 29 subgraphs (*Figure 1B*). In the most extreme case, only one path was accessible from WT to the quadruple mutant WLFA (*Figure 1A*). We also observed a substantial skew in the computed probability of realization among accessible direct paths (*Figure 1—figure supplement 5*), suggesting that most of the realizations in adaptation were captured by a small fraction of possible trajectories (*Weinreich et al., 2006*). These results indicated the existence of sign epistasis and reciprocal sign epistasis, both of which may constrain the accessibility of direct paths (*Weinreich et al., 2006*; *Tufts et al., 2015*). Indeed, we found that these two types of epistasis were prevalent in our fitness landscape (*Figure 1C*). Furthermore, we classified the types of all 24 pairwise epistasis in each subgraph and computed the level of ruggedness as $f_{sign} + 2f_{reciprocal}$, where $f_{type}$ was the fraction of each type of pairwise epistasis. As expected, the number of selectively inaccessible direct paths, i.e. paths that involve fitness declines, was found to be positively correlated with the ruggedness induced by pairwise epistasis (*Figure 1—figure supplement 6*, Pearson correlation = 0.66, p=1.0 × 10$^{-4}$) (*Poelwijk et al., 2007*).

## Two distinct mechanisms of extra-dimensional bypass

Our findings support the view that direct paths of protein adaptation are often constrained by pairwise epistasis on a rugged fitness landscape (*Weinreich et al., 2005*; *Kondrashov and Kondrashov, 2015*). In particular, adaptation can be trapped when direct paths are blocked by reciprocal sign epistasis. However, crucially, this analysis was limited to mutational trajectories within a subgraph of the sequence space. In reality, the dimensionality of protein sequence space is higher. Intuitively, when an extra dimension is introduced, a local maximum may become a saddle point and allow for further adaptation – a phenomenon that is also known as "extra-dimensional bypass" (*Gavrilets, 1997*; *Cariani, 2002*; *Gutiérrez and Maere, 2014*). With our experimental data, we observed two distinct mechanisms of bypass, either using an extra amino acid at the same site or using an additional site, that allow proteins to continue adaptation when no direct paths were accessible due to reciprocal sign epistasis (*Figure 2*). The first mechanism of bypass, which we termed "conversion

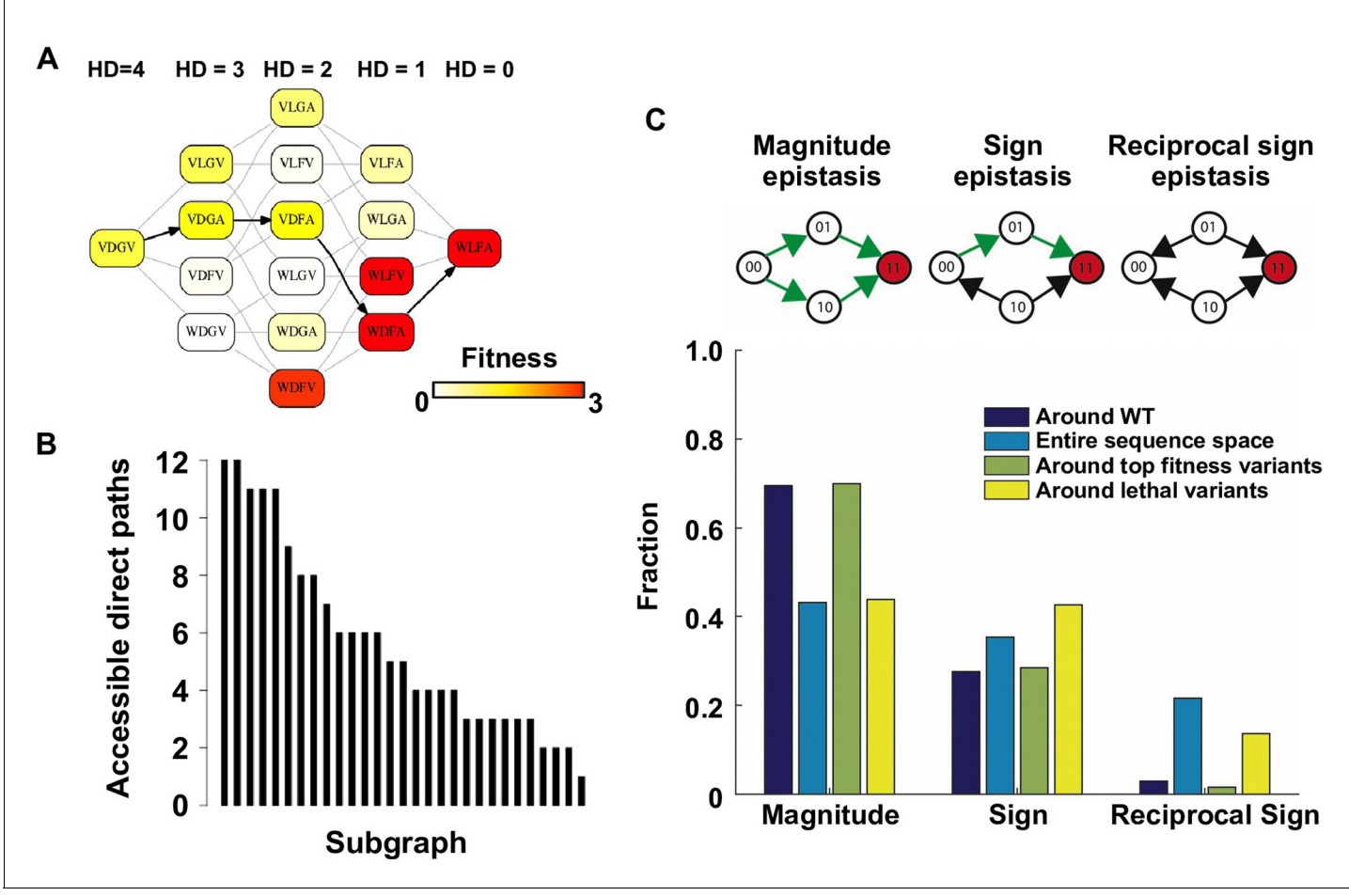

**Figure 1.** Direct paths of adaptation are constrained by pairwise epistasis. (**A**) An example of subgraph that contains VDGV (wild type, WT), the quadruple mutant WLFA and all intermediates between them. Each variant in the subgraph is represented by a node. Edges are drawn between nearest neighbors. The arrows in bold represent the only accessible direct path of adaptation from VDGV to WLFA. HD: Hamming distance. (**B**) We identified a total of 29 subgraphs in which the quadruple mutant was the only fitness peak. The number of accessible direct paths from WT to the quadruple mutant is shown for each subgraph. The maximum number of direct paths is 24. (**C**) The fraction of three types of pairwise epistasis around WT (2091 out of 2166), randomly sampled from the entire sequence space ($10^5$ in total), or in the neighborhood of the top 100 fitness variants and 100 lethal variants. We note that this analysis is different from previous studies on how epistasis changes along adaptive walks, where the quadruples are chosen such that the fitness values of genotype 00, 01 and 11 are in increasing order (*Greene and Crona, 2014*). Sign epistasis and reciprocal sign epistasis, both of which can block adaptive paths, are prevalent in the fitness landscape. Classification scheme of epistasis is shown at the top. Each node represents a genotype, which is within a sequence space of two loci and two alleles. Green arrows represent the accessible paths from genotype "00" to a beneficial double mutant "11" (colored in red).

The following figure supplements are available for figure 1:

**Figure supplement 1.** The four-site sequence space of protein G.

**Figure supplement 2.** Positive epistasis is enriched in the four-site sequence space.

**Figure supplement 3.** Workflow of mRNA display and data validation.

**Figure supplement 4.** Comparison of the wild type to the ensemble of possible genotypes.

**Figure supplement 5.** Subgraph analysis.

**Figure supplement 6.** Correlation between the number of selectively inaccessible direct paths and ruggedness.

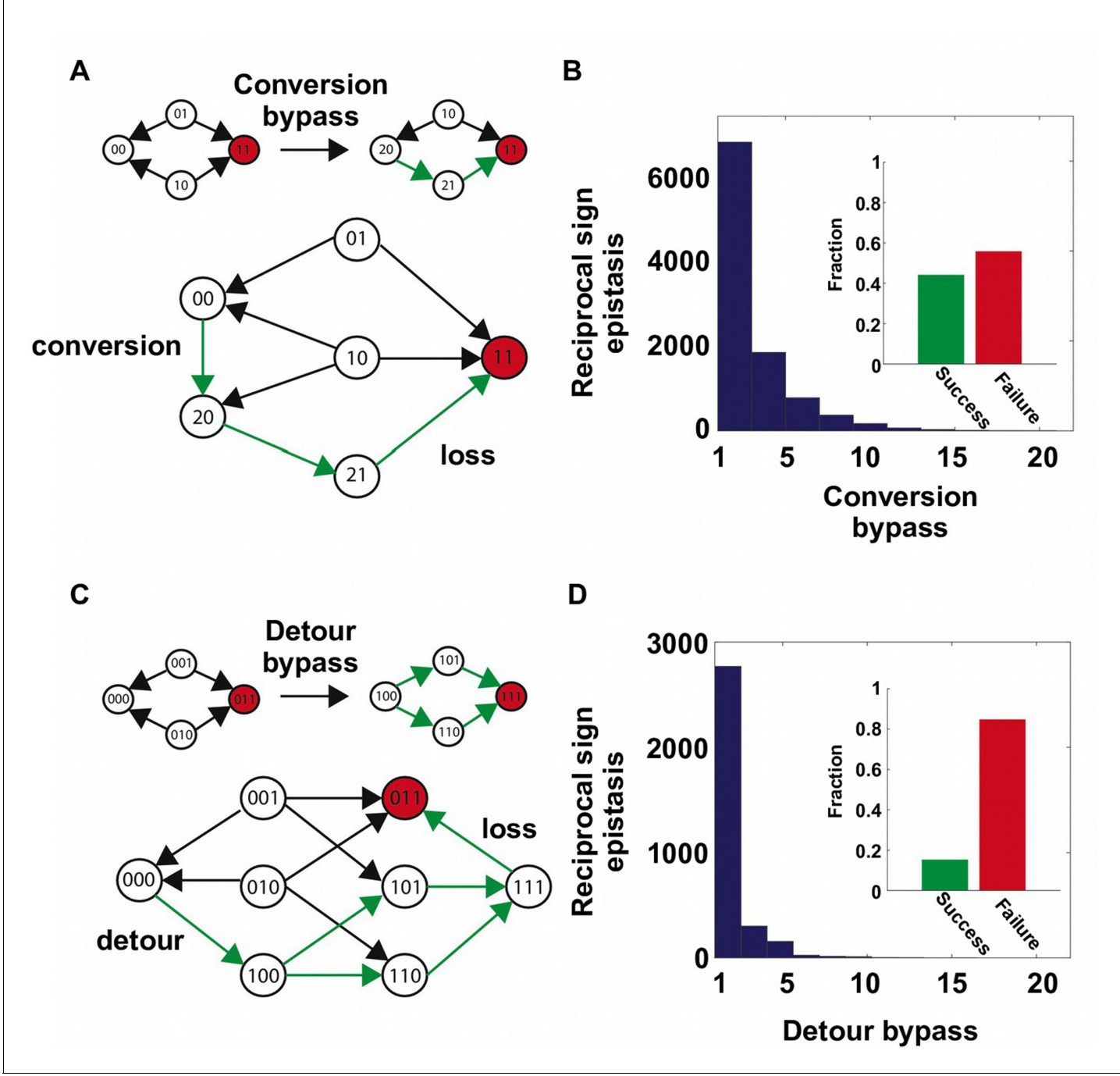

**Figure 2.** Two distinct mechanisms of extra-dimensional bypass. (A) Extra amino acids at one of the two interacting sites may open up potential paths that circumvent the reciprocal sign epistasis. The starting point is 00 and the destination is 11 (in red). Green arrows indicate the accessible path. A successful bypass would require a "conversion" step that substitutes one of the two interacting sites with an extra amino acid (00 → 20), followed by the loss of this mutation later (21 → 11). The original reciprocal sign epistasis is changed to sign epistasis on the new genetic background after conversion. (B) Among ~20,000 randomly sampled reciprocal sign epistasis, >40% of them can be circumvented by at least one conversion bypass (i.e. success, inset). The number of available bypass for the success cases is shown as histogram. (C) The second mechanism of bypass involves an additional site. In this case, adaptation involves a "detour" step to gain mutation at the third site (000 → 100), followed by the loss of this mutation (111 → 011). The original reciprocal sign epistasis is changed to either magnitude epistasis or sign epistasis on the new genetic background after detour (*Figure 2— figure supplement 1*). (D) In comparison to conversion bypass, detour bypass has a lower probability of success (<20%, inset) and is less prevalent.

The following figure supplement is available for figure 2:

*Figure 2 continued on next page*

*Figure 2 continued*

**Figure supplement 1.** Three scenarios of extra-dimensional bypass via an extra site.

bypass", works by converting to an extra amino acid at one of the interacting sites (*Palmer et al., 2015*). Consider a simple scenario with only two interacting sites. If the sequence space is limited to 2 amino acids at each site, as in past analyses of adaptive trajectories, the number of neighbors is 2; however, if all 20 possible amino acids were considered, the total number of neighbors would be 38. Some of these 36 extra neighbors may lead to potential routes that circumvent the reciprocal sign epistasis (*Figure 2A*). In this case, a successful bypass would require a conversion step that substitutes one of the two interacting sites with an extra amino acid ($00 \rightarrow 20$), followed by the loss of this mutation ($21 \rightarrow 11$). This bypass is feasible only if the original reciprocal sign epistasis is changed to sign epistasis after the conversion. To test whether such bypasses were present in our system, we randomly sampled $10^5$ pairwise interactions from the sequence space and analyzed the ~20,000 reciprocal sign epistasis among them (see Materials and methods). More than 40% of the time there was at least one successful conversion bypass and in many cases multiple bypasses were available (*Figure 2B*).

The second mechanism of bypass, which we termed "detour bypass", involves an additional site (*Figure 2C*). In this case, adaptation can proceed by taking a detour step to gain a mutation at the third site ($000 \rightarrow 100$), followed by the later loss of this mutation ($111 \rightarrow 011$) (*DePristo et al., 2007*; *Palmer et al., 2015*). Detour bypass was observed in our system (*Figure 2D*), but was not as prevalent and had a lower probability of success than conversion. Out of 38 possible detour bypasses for a chosen reciprocal sign epistasis, we found that there were on average 1.2 conversion bypasses and 0.27 detour bypasses available. We note, however, that the lower prevalence of detour bypass in our fitness landscape ($L=4$) does not necessarily mean that it should be expected to be less frequent than conversion bypass in other systems. While the maximum number of possible conversion bypasses is always fixed ($19 \times 2 - 2 = 36$), the maximum number of possible detour bypasses ($19 \times (L - 2)$) is proportional to the sequence length $L$ of the entire protein (whereas our study uses a subset $L = 4$). The pervasiveness of extra-dimensional bypasses in our system contrasts with the prevailing view that adaptive evolution is often blocked by reciprocal sign epistasis, when only direct paths of adaptation are considered. The two distinct mechanisms of bypass both require the use of indirect paths, where the Hamming distance to the destination is either unchanged (conversion) or increased (detour).

## Evidence and impacts of higher-order epistasis

In order to circumvent the inaccessible direct paths via extra dimensions, reciprocal sign epistasis must be changed into other types of pairwise epistasis. For detour bypass, this means that the original reciprocal sign epistasis is changed to either magnitude epistasis or sign epistasis in the presence of a third mutation (*Figure 2—figure supplement 1A*). There are three possible scenarios where detour bypass can occur (*Figure 2—figure supplement 1B–D*). We proved that higher-order epistasis is necessary for the scenario that reciprocal sign epistasis is changed to magnitude epistasis, as well as for one of the two scenarios that reciprocal sign epistasis is changed to sign epistasis (see Materials and methods). This suggests a critical role of higher-order epistasis in mediating detour bypass.

To confirm the presence of higher-order epistasis, we decomposed the fitness landscape by Fourier analysis (see Materials and methods, *Figure 3—figure supplement 1*) (*Szendro et al., 2013*; *Weinreich et al., 2013*; *Neidhart et al., 2013*). The Fourier coefficients can be interpreted as epistatic interactions of different orders (*Weinreich et al., 2013*; *de Visser and Krug, 2014*), including the main effects of single mutations (the first order), pairwise epistasis (the second order), and higher-order epistasis (the third and the fourth order). The fitness of variants can be reconstructed by expansion of Fourier coefficients up to a certain order (*Figure 3—figure supplement 2*). In our system with four sites, the fourth order Fourier expansion will always reproduce the measured fitness (i.e. the fraction of variance in fitness explained equals 1). When the second order Fourier expansion does not reproduce the measured fitness, it indicates the presence of higher-order epistasis. In this

way, we identified the 0.1% of subgraphs with greatest fitness contribution from higher-order epistasis (*Figure 3A*, red lines) and visualized the corresponding quadruple mutants by the sequence logo plot (*Figure 3B*). The skewed composition of amino acids in these subgraphs indicates that higher-order interactions are enriched among specific amino acid combinations of site 39, 41 and 54. This interaction among 3 sites is consistent with our knowledge of the protein structure, where the side chains of sites 39, 41, and 54 can physically interact with each other at the core (*Figure 1—figure supplement 1A*) and destabilize the protein due to steric effects (*Figure 3—figure supplement 3*).

In the presence of higher-order epistasis, epistasis between any two sites would vary across different genetic backgrounds. We computed the magnitude of pairwise epistasis ($\varepsilon$) between each pair of amino acid substitutions (see Materials and methods) (*Khan et al., 2011*), and observed numerous instances where the sign of pairwise epistasis depended on genetic background. For example, G41L and V54H were positively epistatic when site 39 was isoleucine [I], but the interaction changed to negative epistasis when site 39 carried a tyrosine [Y] or a tryptophan [W] (*Figure 3C–D*). Similar patterns were observed in other pairwise interactions among site 39, 41 and 54, such as G41F/V54A and V39W/V54H (*Figure 3—figure supplement 4*). The observed pattern of higher-order epistasis was consistent with the results of the Fourier analysis (*Figure 3B*). For example, site 40 was mostly excluded from higher-order epistasis; tyrosine [Y] or tryptophan [W] at site 39 were involved in the most significant higher-order interactions, as they often changed the sign of pairwise epistasis. Higher-order epistasis can also switch the type of pairwise epistasis, such as shifting from reciprocal sign epistasis to magnitude or sign epistasis (*Figure 3—figure supplement 5*), which in turn is important for the existence of detour bypass.

## Indirect paths promote evolutionary accessibility

Our analysis on circumventing reciprocal sign epistasis revealed how indirect paths could open up new avenues of adaptation. To study the impact of indirect paths at a global scale, we performed simulated adaptation in the entire sequence space of 160,000 variants. The fitness landscape was completed by imputing fitness values of the 10,639 missing variants (i.e. 6.6% of the sequence space) that had fewer than 10 sequencing read counts in the input library. Our model of protein fitness incorporated main effects of single mutations, pairwise interactions, and three-way interactions among site 39, 41 and 54 (see Materials and methods, *Figure 4—figure supplement 1*). We used predictor selection based on biological knowledge, followed by regularized regression, which has been demonstrated to ameliorate possible bias in the inferred fitness landscape (*Otwinowski and Plotkin, 2014*). In the complete sequence space, we identified a total of 30 fitness peaks (i.e. local maxima); among them 15 peaks had fitness larger than WT and their combined basins of attraction covered 99% of the sequence space (*Figure 4A*).

We then simulated adaptation on the fitness landscape using three different models of adaptive walks (see Materials and methods), namely the Greedy Model (*de Visser and Krug, 2014*), Correlated Fixation Model (*Gillespie, 1984*), and Equal Fixation Model (*Weinreich et al., 2006*). In the Greedy Model, adaptation proceeds by sequential fixation of mutations that render the largest fitness gain at each step. The other two models assign a nonzero fixation probability to all beneficial mutations, either weighted by (Correlated Fixation Model) or independent of (Equal Fixation Model) the relative fitness gain. The Greedy Model represents adaptive evolution of a large population with pervasive clonal interference (*de Visser and Krug, 2014*). The Correlated Fixation Model represents adaptive evolution of a population under the scheme of strong-selection/weak-mutation (SSWM) (*Gillespie, 1984*), which assumes that the time to fixation is much shorter than the time between mutations and the fixation probability of a given mutation is proportional to the improvement in fitness. The Equal Fixation Model represents a simplified scenario of adaptation where all beneficial mutations fix with equal probability (*Weinreich et al., 2006*).

Among all the possible adaptive paths to fitness peaks, many of them involved indirect paths, i.e. they employed mechanisms of extra-dimensional bypass (*Figure 4B*, *Figure 4—figure supplement 2*). We classified each step on the adaptive paths into three categories based on the change of Hamming distance to the destination (a fitness peak, in this case): "towards (-1)", "conversion (0)", and "detour (+1)" (*Figure 4C*). Conversion was found to be pervasive during adaptation in our fitness landscape (17% of mutational steps for Greedy Model, 41% for Correlated Fixation Model, 59% for Equal Fixation Model). The use of detour was less frequent (0.1% of mutational steps for Greedy Model, 1.3% for Correlated Fixation Model, 3.7% for Equal Fixation Model), in accordance with the

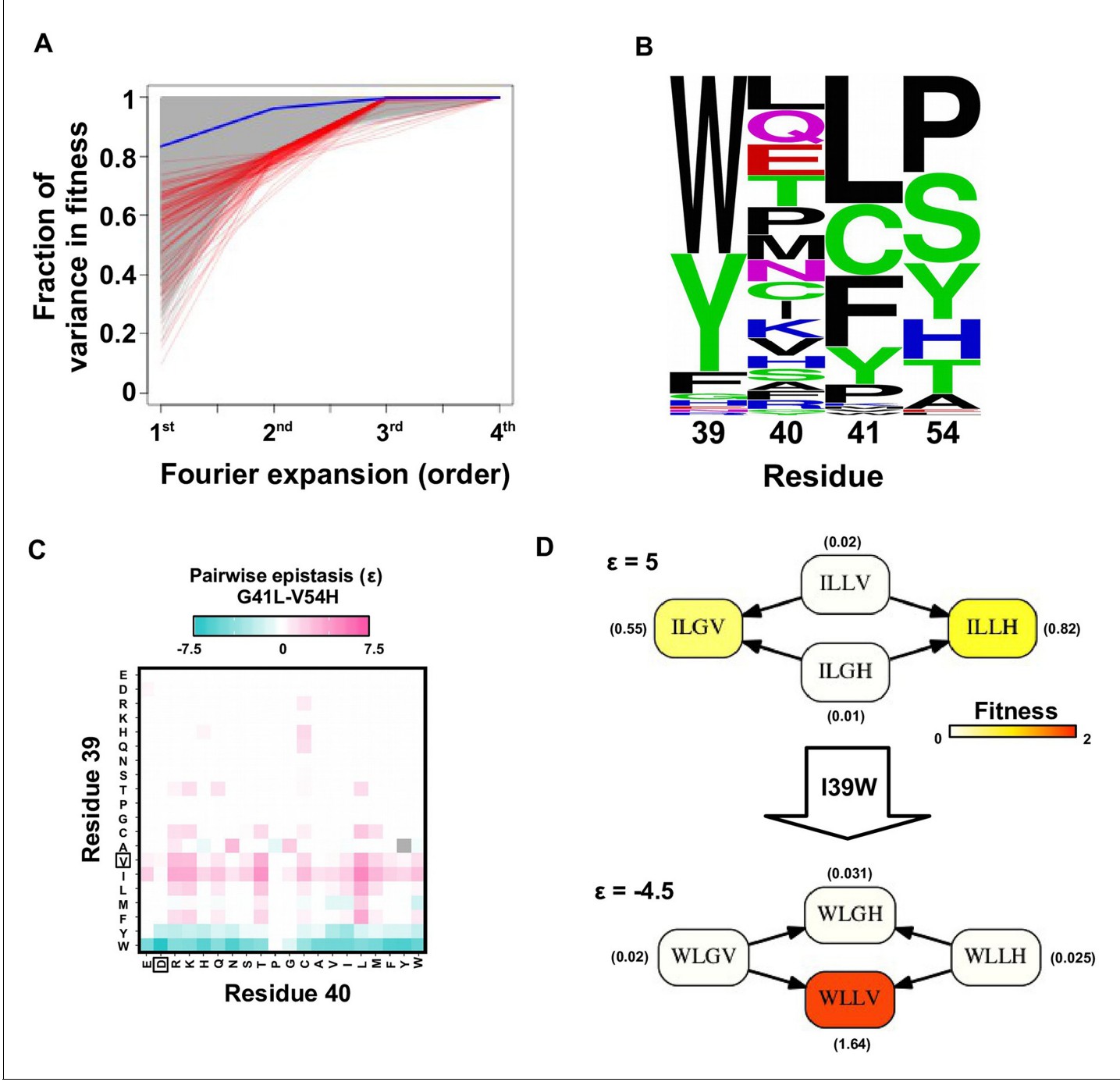

**Figure 3.** Evidence of higher-order epistasis. (A) The fitness decomposition was performed on all subgraphs without missing variants. The fitness of variants can be reconstructed using Fourier coefficients truncated to a certain order. The fraction of variance in fitness explained by expansion of Fourier coefficients truncated to different orders (from first to fourth) is shown for each subgraph. The blue line corresponds to the median. The top 0.1% subgraphs with fitness contributions from higher-order epistasis (the bottom 0.1% subgraphs ranked by fraction of variance explained at second order expansion) are shown in red lines. (B) A sequence logo was generated for the quadruple mutants corresponding to the top 0.1% subgraphs with higher-order epistasis. The skewed composition of amino acids indicates that higher-order interactions are enriched among specific amino acid combinations of site 39, 41 and 54. (C) The magnitude of pairwise epistasis between G41L and V54H across different genetic backgrounds (i.e. all combinations of amino acids at site 39 and 40) is shown as a heat map. The amino acids of WT are boxed. Epistasis that cannot be determined due to missing variant is colored in grey. (D) Altering the genetic background at site 39 changed the positive epistasis ($\varepsilon > 0$) between G41L and V54H to negative epistasis ($\varepsilon < 0$). The fitness of each variant is indicated in the parentheses.

*Figure 3 continued on next page*

*Figure 3 continued*

The following figure supplements are available for figure 3:

**Figure supplement 1.** The fraction of variance explained by Fourier coefficients at each order.

**Figure supplement 2.** Fourier analysis decomposes the fitness landscape into epistatic interactions of different orders.

**Figure supplement 3.** Relationship between fitness, size of the protein core, and predicted $\Delta\Delta G$.

**Figure supplement 4.** Alteration of pairwise epistatic effect under different genetic backgrounds.

**Figure supplement 5.** Higher-order epistasis can change the type of pairwise epistasis.

**Figure supplement 6.** Higher-order epistasis increases the ruggedness of fitness landscapes.

previous observation that detour bypass was less available than conversion bypass in our fitness landscape with $L = 4$. A conversion step would increase the length of an adaptive path by 1, while a detour step would increase the length by 2. As a result, an indirect path can be substantially longer than a direct path consisting of only "towards" steps. We found that many of the adaptive paths required more than 4 steps, which was the maximal length of a direct path between any variants in this landscape (*Figure 4D*). Interestingly, because indirect adaptive paths involved more variants of intermediate fitness, the use of conversion and detour steps depended on the strength of selection. Consistent with previous studies (*Orr, 2002*, *2003*), when mutations conferring larger fitness gains were more likely to fix (e.g. Greedy Model and Correlated Fixation Model), adaptation favored direct moves toward the destination, thus leading to a shorter adaptive paths (*Figure 4C–D*). This suggests that the strength of selection interacts with the topological structure of fitness landscapes to determine the length and directness of evolutionary trajectories.

Given that extra-dimensional bypasses can help proteins avoid evolutionary traps, we expect that their existence would facilitate adaptation in rugged fitness landscapes. Indeed, we found that indirect paths increased the number of genotypes with access to each fitness peak (*Figure 4E*). In addition, the fraction of genotypes with accessible paths to all 15 fitness peaks increased from from 34% to 93% when indirect adaptive paths were allowed (*Figure 4—figure supplement 2C*). We also found that a substantial fraction of beneficial variants (fitness > 1) in the sequence space were accessible from WT only if indirect paths were used (*Figure 4F*). We repeated the analysis in *Figure 4F* with the consideration of the constraints imposed by the standard genetic code (*Figure 4—figure supplement 3A*). The constraints from the genetic code decreased the number of accessible variants due to the reduction in connectivity. However, this reduction in connectivity did not alter our core finding that indirect paths substantially increase evolutionary accessibility (*Figure 4—figure supplement 3B*). Taken together, these results suggest that indirect paths promote evolutionary accessibility in rugged fitness landscapes. This enhanced accessibility would allow proteins to explore more sequence space and lead to delayed commitment to evolutionary fates (i.e. fitness peaks) (*Palmer et al., 2015*). Consistent with this expectation, our simulations showed that many mutational trajectories involving extra-dimensional bypass did not fully commit to a fitness peak until the last two steps (*Figure 4—figure supplement 4*).

## Discussion

In our analysis, we have limited adaptation to the regime where fitness is monotonically increasing via sequential fixation of one-step beneficial mutants. When this assumption is relaxed, adaptation can sometimes proceed by crossing fitness valleys, such as via genetic drift or recombination (*de Visser and Krug, 2014*; *Weissman et al., 2009*; *Ostman et al., 2012*; *Poelwijk et al., 2007*; *Weissman et al., 2010*). Another simplification in most of our analyses is to treat all sequences in a "protein space" (*Smith, 1970*), where two sequences are considered as neighbors if they differ by a single amino-acid substitution. In practice, amino acid substitutions occurring via a single nucleotide mutation are limited by the genetic code, so the total number of one-step neighbors would be

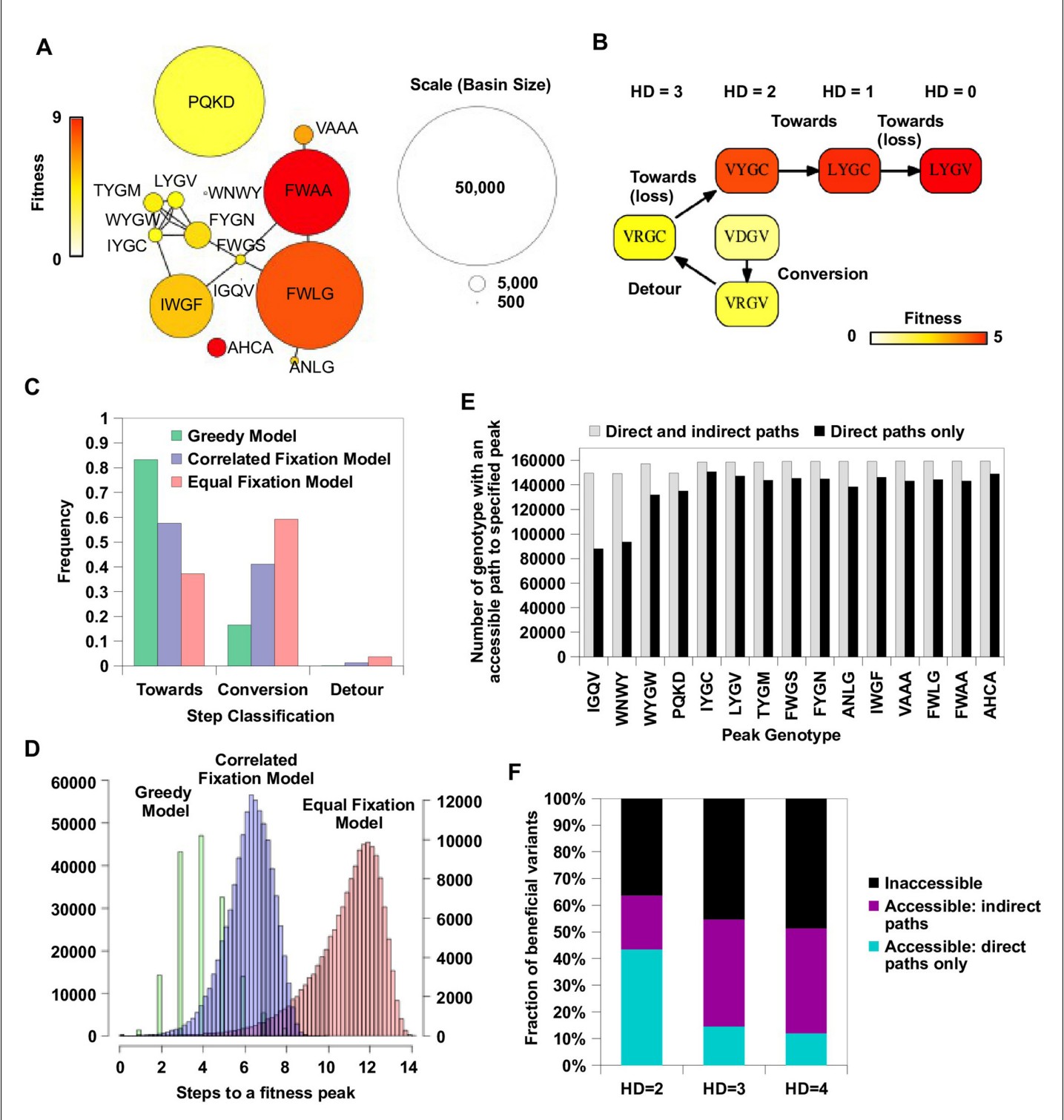

**Figure 4.** Indirect paths promote evolutionary accessibility. (**A**) 15 peaks had fitness larger than WT and their combined basins of attraction accounted for 99% of the entire sequence space. The size of each basin of attraction is identified by the Greedy Model (see Materials and methods). The area of each node is in proportion to the size of the basin of attraction of the corresponding fitness peak. An edge is drawn between fitness peaks that are separated by a Hamming distance of 2. (**B**) A possible adaptive path starting from WT (VDGV) to the fitness peak LYGV. (**C**) The frequency of different types of mutational step are shown. Three models, including the Greedy Model (green), Correlated Fixation Model (blue) and Equal Fixation Model (red), are used to simulate 1000 adaptive paths starting from each variant in the sequence space. All the adaptive paths end at a fitness peak. (**D**) The distribution of the length of the adaptive path initiated at different starting points. For Correlated Fixation Model and Equal Fixation Model, the length

*Figure 4 continued on next page*

*Figure 4 continued*

was computed by averaging over 1000 simulated paths for each starting point. The scale on the left is for Greedy Model. The scale on the right is for Correlated Fixation Model and Equal Fixation Model. (E) Indirect paths increased the number of genotypes accessible to each fitness peak. The 15 peaks are ordered by increasing fitness (from left to right). (F) A large fraction of beneficial variants in the sequence space (fitness > 1) were accessible from WT only via indirect paths. Beneficial variants were classified by their Hamming distance (HD) from WT.

The following figure supplements are available for figure 4:

**Figure supplement 1.** Lasso regression.

**Figure supplement 2.** Indirect paths in adaptation.

**Figure supplement 3.** Evolutionary accessibility under constraints imposed by the standard genetic code.

**Figure supplement 4.** Delay of commitment in mutational trajectories involving extra-dimensional bypass.

reduced from $19L$ to approximately $6L$ (*Figure 4—figure supplement 3*). We also expect fitness landscapes of different systems to have different topological structure. Even in our system (with >93% coverage of the genotype space), the global structure of the fitness landscape is influenced by the imputed fitness values of missing variants, which can vary when different fitness models or fitting methods are used. Our analysis also ignored measurement errors, but the measurement errors are expected to be very small due to the high reproducibility in the data (*Figure 1—figure supplement 3B*). Both imputation of missing variants and measurement errors can lead to slight mis-specification of the topological structure of the fitness landscape. Finally, we note that the four amino acids chosen in our study are in physical proximity and have strong epistatic interactions. While the availability of conversion bypass only depends on the dimensionality at each site, the degree of higher-order epistasis and the availability of detour bypasses can be quite different in other fitness landscapes. Although the details of a particular fitness landscape can influence the quantitative role of different bypass mechanisms, this does not undermine the generality of our conceptual findings on extra-dimensional bypass, higher-order epistasis, and their roles in protein evolution.

Higher-order epistasis has been reported in a few biological systems (*Wang et al., 2013*; *Pettersson et al., 2011*; *Palmer et al., 2015*), and is likely to be common in nature (*Weinreich et al., 2013*). In this study, we observed the presence of higher-order epistasis and systematically quantified its contribution to protein fitness. Our results suggest that higher-order epistasis can either increase or decrease the ruggedness induced by pairwise epistasis, which in turn determines the accessibility of direct paths in a rugged fitness landscape (*Figure 3—figure supplement 6*). We also revealed the important role of higher-order epistasis in mediating detour bypass, which could promote evolutionary accessibility via indirect paths. Our work demonstrates that even in the most rugged regions of a protein fitness landscape, most of the sequence space can remain highly accessible owing to the indirect paths opened up by high dimensionality. The enhanced accessibility mediated by indirect paths may provide a partial explanation for some observations in viral evolution. For example, throughout the course of infection HIV always seems to find a way to revert to the global consensus sequence, a putatively "optimal" HIV-1 sequence after immune invasion (*Zanini et al., 2015*). As we pointed out, the possible number of detour bypasses scales up with sequence length, so it will be interesting to study how extra-dimensional bypass influences adaptation in sequence space of even higher dimensionality. For example, it is plausible that the sequence of a large protein may never be trapped in adaptation (*Gavrilets, 1997*), so that adaptive accessibility becomes a quantitative rather than qualitative problem.

Given the continuing development of sequencing technology, we anticipate that the scale of experimentally determined fitness landscapes will further increase, yet the full protein sequence space is too huge to be mapped exhaustively. Does this mean that we will never be able to understand the full complexity of fitness landscapes? Or perhaps big data from high-throughput measurements will guide us to find general rules? By coupling state-of-the-art experimental techniques with novel quantitative analysis of fitness landscapes, this work takes the optimistic view that we can push

the boundary further and discover new mechanisms underlying evolution (*Fisher et al., 2013*; *Desai, 2013*; *Szendro et al., 2013*)).

## Materials and methods

### Mutant library construction

Two oligonucleotides (Integrated DNA Technologies, Coralville, IA), 5'-AGT CTA GTA TCC AAC GGC NNS NNS NNK GAA TGG ACC TAC GAC GAC GCT ACC AAA ACC TT-3' and 5'-TTG TAA TCG GAT CCT CCG GAT TCG GTM NNC GTG AAG GTT TTG GTA GCG TCG TCG T-3' were annealed by heating to 95°C for 5 min and cooling to room temperature over 1 hr. The annealed nucleotide was extended in a reaction containing 0.5 µM of each oligonucleotide, 50 mM NaCl, 10 mM Tris-HCl pH 7.9, 10 mM MgCl$_2$, 1 mM DTT, 250 µM each dNTP, and 50 units Klenow exo- (New England Biolabs, Ipswich, MA) for 30 mins at 37°C. The product (cassette I) was purified by PureLink PCR Purification Kit (Life Technologies, Carlsbad, CA) according to manufacturer's instructions.

A constant region was generated by PCR amplification using KOD DNA polymerase (EMD Millipore, Billerica, MA) with 1.5 mM MgSO$_4$, 0.2 mM of each dNTP (dATP, dCTP, dGTP, and dTTP), 0.05 ng protein GB1 wild type (WT) template, and 0.5 µM each of 5'-TTC TAA TAC GAC TCA CTA TAG GGA CAA TTA CTA TTT ACA TAT CCA CCA TG-3' and 5'-AGT CTA GTA TCC TCG ACG CCG TTG TCG TTA GCG TAC TGC-3'. The sequence of the WT template consisted of a T7 promoter, 5' UTR, the coding sequence of Protein GB1, 3' poly-GS linkers, and a FLAG-tag (*Figure 1— figure supplement 1B*) ([*Olson et al., 2014*]). The thermocycler was set as follows: 2 min at 95°C, then 18 three-step cycles of 20 s at 95°C, 15 s at 58°C, and 20 s at 68°C, and 1 min final extension at 68°C. The product (constant region) was purified by PureLink PCR Purification Kit (Life Technologies) according to manufacturer's instructions. Both the purified constant region and cassette I were digested with BciVI (New England Biolabs) and purified by PureLink PCR Purification Kit (Life Technologies) according to manufacturer's instructions.

Ligation between the constant region and cassette I (molar ratio of 1:1) was performed using T4 DNA ligase (New England Biolabs). Agarose gel electrophoresis was performed to separate the ligated product from the reactants. The ligated product was purified from the agarose gel using Zymoclean Gel DNA Recovery Kit (Zymo Research, Irvine, CA) according to manufacturer's instructions. PCR amplification was then performed using KOD DNA polymerase (EMD Millipore) with 1.5 mM MgSO$_4$, 0.2 mM of each dNTP (dATP, dCTP, dGTP, and dTTP), 4 ng of the ligated product, and 0.5 µM each of 5'-TTC TAA TAC GAC TCA CTA TAG GGA CAA TTA CTA TTT ACA TAT CCA CCA TG-3' and 5'-GGA GCC GCT ACC CTT ATC GTC GTC ATC CTT GTA ATC GGA TCC TCC GGA TTC-3'. The thermocycler was set as follows: 2 min at 95°C, then 10 three-step cycles of 20 s at 95°C, 15 s at 56°C, and 20 s at 68°C, and 1 min final extension at 68°C. The product, which is referred as "DNA library", was purified by PureLink PCR Purification Kit (Life Technologies) according to manufacturer's instructions.

### Affinity selection by mRNA display

Affinity selection by mRNA display (*Roberts and Szostak, 1997*; *Olson et al., 2012*) was performed as described (*Figure 1—figure supplement 3A*) (*Olson et al., 2014*). Briefly, The DNA library was transcribed by T7 RNA polymerase (Life Technologies) according to manufacturer's instructions. Ligation was performed using 1 nmol of mRNA, 1.1 nmol of 5'-TTT TTT TTT TTT GGA GCC GCT ACC-3', and 1.2 nmol of 5'-/5Phos/-d(A)21-(C$_9$)3-d(ACC)-Puromycin by T4 DNA ligase (New England Biolabs) in a 100 µL reaction. The ligated product was purified by urea PAGE and translated in a 100 µL reaction volume using Retic Lysate IVT Kit (Life Technologies) according to manufacturer's instructions followed by incubation with 500 mM final concentration of KCl and 60 mM final concentration of MgCl$_2$ for at least 30 min at room temperature to increase the efficiency for fusion formation (*Liu et al., 2000*). The mRNA-protein fusion was then purified using ANTI-FLAG M2 Affinity Gel (Sigma-Aldrich, St. Louis, MO). Elution was performed using 3X FLAG peptide (Sigma-Aldrich). The purified mRNA-protein fusion was reverse transcribed using SuperScript III Reverse Transcriptase (Life Technologies). This reverse transcribed product, which was referred as "input library", was incubated with Pierce streptavidin agarose (SA) beads (Life Technologies) that were conjugated with

biotinylated human IgG-FC (Rockland Immunochemicals, Limerick, PA). After washing, the immobilized mRNA-protein fusion was eluted by heating to 95°C. The eluted sample was referred as "selected library".

## Sequencing library preparation

PCR amplification was performed using KOD DNA polymerase (EMD Millipore) with 1.5 mM MgSO$_4$, 0.2 mM of each dNTP (dATP, dCTP, dGTP, and dTTP), the selected library, and 0.5 µM each of 5'-CTA CAC GAC GCT CTT CCG ATC TNN NAG CAG TAC GCT AAC GAC AAC G-3' and 5'-TGC TGA ACC GCT CTT CCG ATC TNN NTA ATC GGA TCC TCC GGA TTC G-3'. The underlined "NNN" indicated the position of the multiplex identifier, GTG for input library and TGT for post-selection library. The thermocycler was set as follows: 2 min at 95°C, then 10 to 12 three-step cycles of 20 s at 95°C, 15 s at 56°C, and 20 s at 68°C, and 1 min final extension at 68°C. The product was then PCR amplified again using KOD DNA polymerase (EMD Millipore) with 1.5 mM MgSO$_4$, 0.2 mM of each dNTP (dATP, dCTP, dGTP, and dTTP), the eluted product from mRNA display, and 0.5 µM each of 5'-AAT GAT ACG GCG ACC ACC GAG ATC TA CAC TCT TTC CCT ACA CGA CGC TCT TCC G-3' and 5'-CAA GCA GAA GAC GGC ATA CGA GAT CGG TCT CGG CAT CCT GCT GA ACC GCT CTT CCG-3'. The thermocycler was set as follows: 2 min at 95°C, then 10 to 12 three-step cycles of 20 s at 95°C, 15 s at 56°C, and 20 s at 68°C, and 1 min final extension at 68°C. The PCR product was then subjected to 2 × 100 bp paired-end sequencing using Illumina HiSeq 2500 platform. We aimed to obtain at least 20 million paired-end reads for each input library and post-selection library such that the average coverage for each variant would be more than 100 paired-end reads. There were 89,075,246 paired-end reads obtained for the input library and 45,587,128 paired-end reads obtained for the post-selection library. Raw sequencing data have been submitted to the NIH Short Read Archive under accession number: BioProject PRJNA278685.

We were able to compute the fitness for 93.4% of all variants from the sequencing data. The fitness measurements in this study were highly consistent with our previous study on the fitness of single and double mutants in protein GB1 (Pearson correlation = 0.97, *Figure 1—figure supplement 3B*) (*Olson et al., 2014*).

## Sequencing data analysis

The first three nucleotides of both forward read and reverse read were used for demultiplexing. If the first three nucleotides of the forward read were different from that of the reverse read, the given paired-end read would be discarded. For both forward read and reverse read, the nucleotides that were corresponding to the codons of protein GB1 sites 39, 40, 41, and 54 were extracted. If coding sequence of sites 39, 40, 41, and 54 in the forward read and that in the reverse read did not reverse-complement each other, the paired-end read would be discarded. Subsequently, the occurrence of individual variants at the amino acid level for site 39, 40, 41, and 54 in both input library and selected library were counted, with each paired-end read represented 1 count. Custom python scripts and bash scripts were used for sequencing data processing. All scripts have been deposited to https://github.com/wchnicholas/ProteinGFourMutants.

## Calculation of fitness

The fitness ($w$) for a given variant $i$ was computed as:

$$w_i = \frac{count_{i,selected}/count_{i,input}}{count_{WT,selected}/count_{WT,input}} \tag{1}$$

where $count_{i,selected}$ represented the count of variant $i$ in the selected library, $count_{i,input}$ represented the count of variant $i$ in the input library, $count_{WT,selected}$ represented the count of WT (VDGV) in the selected library, and $count_{WT,input}$ represented the count of WT (VDGV) in the input library.

Therefore, the fitness of each variant, $w_i$, could be viewed as the fitness relative to WT (VDGV), such that = 1. Variants with $count_{input} < 10$ were filtered to reduce noise. The fraction of all possible variants that passed this filter was 93.4% (149,361 out of 160,000 all possible variants).

The fitness of each single substitution variant was referenced to our previous study (*Olson et al., 2014*), because the sequencing coverage of single substitution variants in our previous study was much higher than in this study (~100 fold higher). Hence, our confidence in computing fitness for a

single substitution variant should also be much higher in our previous study than this study. Subsequently, the fitness of each single substitution in this study was calculated by multiplying a factor of 1.159 by the fitness of that single substitution computed from our previous study (*Olson et al., 2014*). This is based on the linear regression analysis between the single substitution fitness as measured in our previous study and in this study, which had a slope of 1.159 and a y-intercept of ~0. The fitness of each profiled variant is shown in *Supplementary file 1*.

## Magnitude and type of pairwise epistasis

The three types of pairwise epistasis (magnitude, sign and reciprocal sign) were classified by ranking the fitness of the four variants involved (*Greene and Crona, 2014*).

To quantify the magnitude of epistasis ($\varepsilon$) between substitutions *a* and *b* on a given background variant *BG*, the relative epistasis model (*Khan et al., 2011*) was employed as follows:

$$\varepsilon_{ab,BG} = \ln(\frac{w_{ab}}{w_{BG}}) - \ln(\frac{w_a}{w_{BG}}) - \ln(\frac{w_b}{w_{BG}}) \tag{2}$$

where $w_{ab}$ represents the fitness of the double substitution, $\ln(w_a)$ and $\ln(w_b)$ represents the fitness of each of the single substitution respectively, and $w_{BG}$ represents the fitness of the background variant.

As described previously (*Olson et al., 2014*), there exists a limitation in determining the exact fitness for very low-fitness variants in this system. To account for this limitation, several rules were adapted from our previous study to minimize potential artifacts in determining $\varepsilon$ (*Olson et al., 2014*). We previously determined that the detection limit of fitness (*w*) in this system is ~0.01 (*Olson et al., 2014*).

Rule 1) if $\max(\frac{w_{ab}}{w_{BG}}, \frac{w_a}{w_{BG}}, \frac{w_b}{w_{BG}}) < 0.01$, $\varepsilon_{ab,BG,adjusted} = 0$

Rule 2) if $\min(w_a, w_b, \frac{w_a}{w_{BG}}, \frac{w_b}{w_{BG}}) < 0.01$, $\varepsilon_{ab,BG,adjusted} = \max(0, \varepsilon_{ab,BG})$

Rule 3) if $\min(w_{ab}, \frac{w_{ab}}{w_{BG}}) < 0.01$, $\varepsilon_{ab,BG,adjusted} = \min(0, \varepsilon_{ab,BG})$

Rule 1 prevents epistasis being artificially estimated from low-fitness variants. Rule 2 prevents overestimation of epistasis due to low fitness of one of the two single substitutions. Rule 3 prevents underestimation of epistasis due to low fitness of the double substitution. Of note, $\varepsilon_{ab,BG,adjusted}$ was set to 0 if both Rule 2 and Rule 3 were satisfied. To compute the epistasis between two substitutions, *a* and *b*, on a given background variant *BG*, $\varepsilon_{ab,BG,adjusted}$ would be used if any one of the above three rules was satisfied. Otherwise, $\varepsilon_{ab,BG}$ would be used.

## Fourier analysis

Fitness decomposition was performed on all subgraphs without missing variants (109,235 subgraphs in total). We decomposed the fitness landscape into epistatic interactions of different orders by Fourier analysis (*Stadler, 1996*; *Szendro et al., 2013*; *Weinreich et al., 2013*; *Neidhart et al., 2013*). The Fourier coefficients given by the transform can be interpreted as epistasis of different orders (*Weinreich et al., 2013*; *de Visser and Krug, 2014*).

For a binary sequence $\vec{z}$ with dimension $L$ ($z_i$ equals 1 if mutation is present at position *i*, or 0 otherwise), the Fourier decomposition theorem states that the fitness function $f(\vec{z})$ can be expressed as (*Weinberger, 1991*):

$$f(\vec{z}) = \sum_{\vec{k}} \hat{f}_{\vec{k}} (-1)^{\vec{z} \cdot \vec{k}} \tag{3}$$

The formula for the Fourier coefficients $\hat{f}_{\vec{k}}$ is then:

$$\hat{f}_{\vec{k}} = \frac{1}{2^L} \sum_{\vec{z}} f(\vec{z}) (-1)^{\vec{z} \cdot \vec{k}} \tag{4}$$

For example, we can expand the fitness landscape up to the second order, i.e. with linear and quadratic terms

$$f(\vec{\sigma}) = \hat{f}_0 + \sum_i \hat{f}_{\vec{e}_i} \sigma_i + \sum_{i<j} \hat{f}_{\vec{e}_i + \vec{e}_j} \sigma_i \sigma_j + \cdots \tag{5}$$

where $\sigma_i \equiv (-1)^{z_i} \in \{+1, -1\}$, and $\vec{e}_i$ is a unit vector along the $i^{th}$ direction. In our analysis of subgraphs, there are a total of $2^4 = 16$ terms in the Fourier decomposition, with $\binom{4}{i}$ terms for the $i^{th}$ order ($i = 0, 1, 2, 3, 4$). We can expand the fitness landscape up to a given order by ignoring all higher-order terms in *Equation 3*. In this paper, we refer to higher-order epistasis as non-zero contribution to fitness from the third order terms and beyond.

## Imputing the fitness of missing variants

The fitness values for 10,639 variants (6.6% of the entire sequence space) were not directly measured (read count in the input pool = 0) or were filtered out because of low read counts in the input pool (see section "Calculation of fitness"). To impute the fitness of these missing variants, we performed regularized regression on fitness values of observed variants using the following model (*Hinkley et al., 2011*; *Otwinowski and Plotkin, 2014*):

$$log(f) = \alpha_0 + \sum_{i=1}^{N_M} \beta_i M_i + \sum_{j=1}^{N_P} \gamma_j P_j + \sum_{k=1}^{N_T} \delta_k T_k \qquad (6)$$

Here, $f$ is the protein fitness. $\alpha_0$ is the intercept that represents the log fitness of WT; $\beta_i$ represents the main effect of a single mutation, $i$; $M_i$ is a dummy variable that equals 1 if the single mutation $i$ is present in the sequence, or 0 if the single mutation is absent; and $N_M = 19 \times \binom{4}{1} = 76$ is the total number of single mutations. Similarly, $\gamma_j$ represents the effect of interaction between a pair of mutations; $P_j$ is the dummy variable that equals either 1 or 0 depending on the presence of that those two mutations; and $N_P = 19^2 \times \binom{4}{2} = 2166$ is the total number of possible pairwise interactions. In addition to the main effects of single mutations and pairwise interactions, the three-way interactions among sites 39, 41 and 54 are included in the model, based on our knowledge of higher-order epistasis (*Figure 3*). $\delta_k$ represents the effect of three-way interactions among sites 39, 41 and 54; $T_k$ is the dummy variable that equals either 1 or 0 depending on the presence of that three-way interaction; and $N_T = 19^3 = 6859$ is the total number of three-way interactions. Thus, the total number of coefficients in this model is 9102, including main effects of each site (i.e. additive effects), interactions between pairs of sites (i.e. pairwise epistasis), and a subset of three-way interactions (i.e. higher-order epistasis).

Out of the 149,361 variants with experimentally measured fitness values, 119,884 variants were non-lethal ($f > 0$) and were used to fit the model coefficients using lasso regression (Matlab R2014b). Lasso regression adds a penalty term $\lambda \sum |\theta|$ ($\theta$ stands for any coefficient in the model) when minimizing the least squares, thus it favors sparse solutions of coefficients (*Figure 4—figure supplement 1B*). We calculated the 10-fold cross-validation MSE (mean squared errors) of the lasso regression for a wide range of penalty parameter $\lambda$ (*Figure 4—figure supplement 1A*). $\lambda = 10^{-4}$ is chosen. For measured variants, the model-predicted fitness values were highly correlated with the actual fitness values (Pearson correlation=0.93, *Figure 4—figure supplement 1C*). We then used the fitted model to impute the fitness of the 10,639 missing variants and complete the entire fitness landscape. Imputed fitness values for missing variants are listed in *Supplementary file 2*.

## Simulating adaptation using three models for fixation

Python package "networkx" was employed to construct a directed graph that represented the entire fitness landscape for sites 39, 40, 41, and 54. A total of $20^4 = 160,000$ nodes were present in the directed graph, where each node represented a 4-site variant. For all pairs of variants separated by a Hamming distance of 1, a directed edge was generated from the variant with a lower fitness to the variant with a higher fitness. Therefore, all successors of a given node had a higher fitness than the given node. A fitness peak was defined as a node that had 0 out-degree. Three models, namely the Greedy Model (*de Visser and Krug, 2014*), the Correlated Fixation Model (*Gillespie, 1984*), and the Equal Fixation Model (*Weinreich et al., 2006*), were employed in this study to simulate the mutational steps in adaptive trajectories. Under all three models, the probability of fixation of a deleterious or neutral mutation is 0. Considering a mutational trajectory initiated at a node, $n_i$ with a fitness value of $w_i$, where $n_i$ has M successors, ($n_1, n_2, \ldots n_M$) with fitness values of ($w_1, w_2, \ldots w_M$). Then the probability that the next mutational step is from $n_i$ to $n_k$, where $k \in (1, 2, \ldots M)$, is denoted $P_{i \rightarrow k}$ and called the probability of fixation, and can be computed for each model as follows.

For the Greedy Model (deterministic model),

$$\text{if } w_k = max(w_1, w_2, ... w_M), P_{i \to k} = 1 \tag{7}$$

$$\text{otherwise, } P_{i \to k} = 0 \tag{8}$$

For the Correlated Fixation Model (non-deterministic model),

$$P_{i \to k} = \frac{w_k - w_i}{\sum_{n=1}^{M}(w_n - w_i)} \tag{9}$$

For the Equal Fixation Model (non-deterministic model),

$$P_{i \to k} = \frac{1}{M} \tag{10}$$

To compute the shortest path from a given variant to all reachable variants, the function "single-source_shortest_path" in "networkx" was used. If the shortest path between a low-fitness variant and a high-fitness variant does not exist, it means that the high-fitness variant is inaccessible. If the length of the shortest path is larger than the Hamming distance between two variants, it means that adaptation requires indirect paths.

Under constraints imposed by the standard genetic code, the connectivity of the directed graph that represented the fitness landscape was restricted according to the matrix shown in *Figure 4—figure supplement 3A*. The genetic distance between two variants was calculated according to the matrix shown in *Figure 4—figure supplement 3A*. If the length of the shortest path is larger than the genetic distance between two variants, it means that adaptation requires indirect paths.

## Analysis of direct paths within a subgraph

In the subgraph analysis shown in *Figure 1—figure supplement 4*, the fitness landscape was restricted to 2 amino acids at each of the 4 sites (the WT and adapted alleles). There was a total of $2^4$ variants, hence nodes, in a given subgraph. Only those subgraphs where the fitness of all variants was measured directly were used (i.e. any subgraph with missing variants was excluded from this analysis). Mutational trajectories were generated in the same manner as in the analysis of the entire fitness landscape (see subsection "Simulating adaptation using three models for fixation"). In a subgraph with only one fitness peak, the probability of a mutational trajectory from node $i$ to node $j$ via intermediate $a$, $b$, and $c$ was as follows:

$$P_{i \to a \to b \to c \to j} = P_{i \to a} \times P_{a \to b} \times P_{b \to c} \times P_{c \to j} \tag{11}$$

To compute the Gini index for a given set of mutational trajectories from node $i$ to node $j$, the probabilities of all possible mutational trajectories were sorted from large to small. Inaccessible trajectories were also included in this sorted list with a probability of 0. This sorted list with $t$ trajectories was denoted as $(P_{i \to j,1}, P_{i \to j,2}, ... P_{i \to j,t})$, where $P_{i \to j,1}$ was the largest and $P_{i \to j,t}$ was the smallest. This sorted list was converted into a list of cumulative probabilities, which is denoted as

$(A_{i \to j,1}, A_{i \to j,2}, ... A_{i \to j,t})$, where $A_{i \to j,t} = \sum_{n=1}^{t} P_{i \to j,t}$.

The Gini index for the given subgraph was then computed as follows:

$$\text{Gini index} = \frac{2 \times \sum_{n=1}^{t-1}(A_{i \to j,n}) + A_{i \to j,t} - t}{t - 1} \tag{12}$$

## Visualization

Sequence logo was generated by WebLogo (http://weblogo.berkeley.edu/logo.cgi) (*Crooks et al., 2004*). The visualization of basins of attraction (*Figure 4A*) was generated using Graphviz with "fdp" as the option for layout.

## ΔΔG prediction

The ΔΔG prediction was performed by the ddg_monomer application in Rosetta software (*Das and Baker, 2008*) using the parameters from row 16 of Table I in Kellogg et al. (*Kellogg et al., 2011*).

## Extra-dimensional bypass is mediated by higher-order epistasis

Here we prove that higher-order epistasis is required for two possible scenarios of extra-dimensional bypass via an additional site (*Figure 2—figure supplement 1*). For a fitness landscape defined on a Boolean hypercube, we can expand the fitness as Taylor series (*Weinberger, 1991*).

$$
\begin{aligned}
f_{000} &= \alpha_0 \\
f_{001} &= \alpha_0 + \alpha_1 \\
f_{010} &= \alpha_0 + \alpha_2 \\
f_{100} &= \alpha_0 + \alpha_3 \\
f_{011} &= \alpha_0 + \alpha_1 + \alpha_2 + \alpha_{12} \\
f_{101} &= \alpha_0 + \alpha_1 + \alpha_3 + \alpha_{13} \\
f_{110} &= \alpha_0 + \alpha_2 + \alpha_3 + \alpha_{23} \\
f_{111} &= \alpha_0 + \alpha_1 + \alpha_2 + \alpha_3 + \alpha_{12} + \alpha_{13} + \alpha_{23} + \alpha_{123}
\end{aligned}
\tag{13}
$$

To prove that higher-order epistasis is present is equivalent to prove that $\alpha_{123} \neq 0$. The fitness difference between neighbors is visualized by the directed edges that go from low-fitness variant to high-fitness variant, thus each edge represents an inequality. No cyclic paths are allowed in this directed graph.

The reciprocal sign epistasis (*Figure 2—figure supplement 1A*) gives,

$$
000 \leftarrow 001 : \alpha_1 < 0 \tag{14}
$$

$$
000 \leftarrow 010 : \alpha_2 < 0 \tag{15}
$$

$$
001 \rightarrow 011 : \alpha_2 + \alpha_{12} > 0 \tag{16}
$$

$$
010 \rightarrow 011 : \alpha_1 + \alpha_{12} > 0 \tag{17}
$$

The detour step ($000 \rightarrow 100$) and the loss step ($111 \rightarrow 011$) are required for extra-dimensional bypass,

$$
000 \rightarrow 100 : \alpha_3 > 0 \tag{18}
$$

$$
011 \leftarrow 111 : \alpha_3 + \alpha_{13} + \alpha_{23} + \alpha_{123} < 0 \tag{19}
$$

For the remaining 6 edges, there are 3 possible configurations (*Figure 2—figure supplement 1B–D*). For the scenario illustrated in (**B**), we have

$$
100 \rightarrow 101 : \alpha_1 + \alpha_{13} > 0 \tag{20}
$$

$$
100 \rightarrow 110 : \alpha_2 + \alpha_{23} > 0 \tag{21}
$$

Combining inequality (14) and (20) gives

$$
\alpha_{13} > 0 \tag{22}
$$

Combining inequality (15) and (21) gives

$$
\alpha_{23} > 0 \tag{23}
$$

Combining the above two inequalities with (18) and (19), we arrive at

$$\alpha_{123} < 0 \tag{24}$$

For the scenario in (**C**), the proof of higher-order epistasis is similar. We have (the yellow edge)

$$001 \rightarrow 101 : \alpha_3 + \alpha_{13} > 0 \tag{25}$$

Combining the above inequality with (15), (19) and (21), we arrive at

$$\alpha_{123} < 0 \tag{26}$$

For the scenario in (**D**), when $\alpha_3 + \alpha_{13} < 0$, all the inequalities can be satisfied with $\alpha_{123} = 0$. So higher-order epistasis is not necessary in this case.

## Acknowledgements

We would like to thank Jesse Bloom and Joshua Plotkin for helpful comments on early versions of the manuscript. NCW was supported by Philip Whitcome Pre-Doctoral Fellowship, Audree Fowler Fellowship in Protein Science, and UCLA Dissertation Year Fellowship. LD was supported by HHMI Postdoctoral Fellowship from Jane Coffin Childs Memorial Fund for Medical Research. RS was supported by NIH R01 DE023591. The funders had no role in study design, data collection and analysis, decision to publish, or preparation of the manuscript.

## Additional information

### Funding

| Funder | Grant reference number | Author |
| --- | --- | --- |
| University of California, Los Angeles | Philip Whitcome Pre-Doctoral Fellowship | Nicholas C Wu |
| Jane Coffin Childs Memorial Fund for Medical Research | HHMI Postdoctoral Fellowship from Jane Coffin Childs Memorial Fund for Medical Research | Lei Dai |
| National Institutes of Health | R01 DE023591 | Ren Sun |
| University of California, Los Angeles | Audree Fowler Fellowship in Protein Science | Nicholas C Wu |
| University of California, Los Angeles | Dissertation Year Fellowship | Nicholas C Wu |

The funders had no role in study design, data collection and interpretation, or the decision to submit the work for publication.

### Author contributions

NCW, Conception and design, Acquisition of data, Analysis and interpretation of data, Drafting or revising the article; LD, JOL-S, Analysis and interpretation of data, Drafting or revising the article; CAO, Conception and design, Acquisition of data; RS, Conception and design, Drafting or revising the article

### Author ORCIDs

Nicholas C Wu, http://orcid.org/0000-0002-9078-6697
James O Lloyd-Smith, http://orcid.org/0000-0001-7941-502X

## Additional files

### Supplementary files

• Supplementary file 1. The fitness of each profiled variant.

• Supplementary file 2. Imputed fitness values for missing variants.

**Major datasets**

The following dataset was generated:

| Author(s) | Year | Dataset title | Dataset URL | Database, license, and accessibility information |
|---|---|---|---|---|
| Wu NC, Dai L, Olson CA, Lloyd-Smith JO, Sun R | 2015 | Streptococcus dysgalactiae strain: GB1_AO | https://www.ncbi.nlm.nih.gov/bioproject/PRJNA278685/ | Publicly available at the NCBI Gene Expression Omnibus (Accession no: PRJNA278685) |

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
