## [Decision Letter]

Thank you for submitting your article "Adaptation in protein fitness landscapes is facilitated by indirect paths" for consideration by *eLife*. Your article has been favorably evaluated by Diethard Tautz as the Senior editor and three reviewers, including Joachim Krug (Reviewer #2) and a member of our Board of Reviewing Editors.

The reviewers have discussed the reviews with one another and the Reviewing Editor has drafted this decision to help you prepare a revised submission.

Summary:

Wu et al. report fitness measurements of all possible of combinations of amino acid substitutions at 4 positions in the protein GB1. Properties of fitness or folding landscapes of proteins are fundamental determinants of evolution since they determine which mutations are permissible on which genetic background. The major contribution of this manuscript is that it for the first time measures all combinations of all amino acids at multiple (four) sites. Using this data, the authors demonstrate that indirect path often enable ascend to the global fitness maximum, even when direct path a blocked by fitness valleys.

Essential revisions:

All reviews agreed that the results reported are a significant step forward in understanding the landscapes governing evolution of proteins. Before we can recommend publication, we would like to see the following points addressed.

1) A more thorough discussion of the generality of the results. The sites were chosen to be as epistatic as possible. Hence it is not clear how representative the properties of the 4 site landscape are. Along the same lines, we would like know how the wild type genotype compares to the ensemble of possible genotypes. Where does the wild type fall in the distribution of all fitness measurements? It would be informative to have Figure 1 not only for WT and the average sequence, but also the top and bottom 5% or similar.

2) Quantification of ruggedness and the importance of higher order coefficients. Figure 3 quantifies the accuracy of the approximation of the fitness landscapes (2 allele subspaces) after inclusion of 1st, 2nd, and 3rd order effects. This is important and useful, but can be improved. Instead of plotting the correlation coefficient, plot the fraction of variance explained by each order (that is the power spectrum of the Fourier decomposition, see Neher and Shraiman, RMP, 2011 or Neidhart et al., JTB 2013). Furthermore, a distribution of the 1st, 2nd, and 3rd order variance would be easier to parse than the current overlay of many lines. Along similar lines, can one quantify whether higher-order epistasis makes the landscape more or less rugged on average? If all higher order coefficients are set to 0, does the number of accessible path go up or down?

3) Your simulations allow transitions from every amino acid to any other amino acid and hence ignore the constraints imposed by the encoding as codons. It does not seem to be a big complication to restrict transitions to those that can be achieved by single nucleotide substitutions. Does this reduction in connectivity reduce the effects of indirect path substantially?

4) Missing literature and inaccurate statements. Even if there are only two amino acids (or more generally two alleles) per site, there can still be indirect paths with mutational reversions. The statements in the second paragraph of the Main text are not accurate. The mechanisms of bypass and conversion have been discussed previously (and have not been "discovered" by the authors; Main text, fifth paragraph). Gavrilets (1997) discusses extra-dimensional bypass. Several recent papers have theoretically studied the effect of reversions on evolutionary accessibility in diallelic sequence spaces:

Julien Berestycki, Eric Brunet, Zhan Shi. http://arxiv.org/abs/1401.6894

Anders Martinsson. http://arxiv.org/abs/1501.0220

Li Li. http://arxiv.org/abs/1502.07642

The adaptation schemes referred to by the authors as 'Greedy model', 'Correlated Fixation Model' and 'Equal Fixation Model' should be placed into their proper population-genetic context. Key references are:

H. Allen Orr. Evolution, 56(7), 2002, pp. 1317-1330

H. Allen Orr. J. theor. Biol. (2003) 220, 241-247

which discuss the performance of the three types of 'adaptive walks' in the uncorrelated random 'mutational landscape' model and show, in particular, that greedy walks are much shorter than 'correlated fixation' walks which in turn are shorter than 'equal fixation' walks. This is expected to be a fairly general pattern that also appears in the results shown in Figure 4.

5) One take home message of the manuscript is that even in the most rugged place of a protein fitness landscape, the multitude of possible mutations makes most places accessible (via detours at times). This could be a way to reconcile the seemingly contradictory observations that (i) there is a lot of epistasis/ruggedness and (ii) that amino acid preferences are preserved (work by Bloom et al) and that viruses like HIV extensively revert to a putatively optimal fitness peak after immune evasion. In high dimensions, fitness landscapes seem to be locally rugged and accessible at the same time. A more thorough discussion of the potentially wider implications of indirect path could place these results into a broader context.

---

## [Author Response]

*Essential revisions:*

*All reviews agreed that the results reported are a significant step forward in understanding the landscapes governing evolution of proteins. Before we can recommend publication, we would like to see the following points addressed.*

1) A more thorough discussion of the generality of the results. The sites were chosen to be as epistatic as possible. Hence it is not clear how representative the properties of the 4 site landscape are. Along the same lines, we would like know how the wild type genotype compares to the ensemble of possible genotypes. Where does the wild type fall in the distribution of all fitness measurements? It would be informative to have Figure 1 not only for WT and the average sequence, but also the top and bottom 5% or similar.

We appreciate the reviewers’ comment on these issues. In the revised manuscript, we have expanded the discussion on the generality of our results. In particular, we have pointed out that strong epistasis among the four sites we investigated would not influence our major findings on the indirect paths: “Finally, we note that the four amino acids chosen in our study are in physical proximity and have strong epistatic interactions. (…) Although the details of a particular fitness landscape can influence the quantitative role of different bypass mechanisms, this does not undermine the generality of our conceptual findings on extra-dimensional bypass, higher-order epistasis, and their roles in protein evolution.”

Based on the reviewers’ comments, we have performed additional analyses to compare the wild type to the ensemble of possible genotypes. The fitness rank of the wild type genotype is 2.4% in the distribution of all fitness measurements. The distributions of fitness values of all non-lethal genotypes are shown in Figure 1—figure supplement 4. Also, as suggested, we have also modified Figure 1 to include analyses on pairwise epistasis in the neighborhood of sequences with fitness values at the top and the bottom of the distribution. Although we used the wild type genotype in illustrations of adaptive pathways (Figure 1 and Figure 4), we want to emphasize that there is nothing unique about the wild type genotype in our analyses of the fitness landscape.

2) Quantification of ruggedness and the importance of higher order coefficients. Figure 3 quantifies the accuracy of the approximation of the fitness landscapes (2 allele subspaces) after inclusion of 1st, 2nd, and 3rd order effects. This is important and useful, but can be improved. Instead of plotting the correlation coefficient, plot the fraction of variance explained by each order (that is the power spectrum of the Fourier decomposition, see Neher and Shraiman, RMP, 2011 or Neidhart et al., JTB 2013). Furthermore, a distribution of the 1st, 2nd, and 3rd order variance would be easier to parse than the current overlay of many lines. Along similar lines, can one quantify whether higher-order epistasis makes the landscape more or less rugged on average? If all higher order coefficients are set to 0, does the number of accessible path go up or down?

We appreciate these suggestions on how to improve the presentation of Fourier coefficients, and we have implemented them in the revised manuscript. Instead of plotting the Pearson correlation, we have updated Figure 3 to show the fraction of variance explained by expanding the Fourier coefficients to 1st, 2nd, 3rd, and 4th order. In addition, we have shown the distribution of the variance in fitness at each order in Figure 3—figure supplement 1.

We have altered the text accordingly: “For the 109,235 complete subgraphs that we analyzed, the fraction of the variance in fitness explained by each order is shown. Although the first order (main effects) and the second order Fourier coefficients (pairwise epistasis) can explain most of the variance in fitness, higher-order epistasis is present in certain combination of mutations.”

To further study the effect of higher-order epistasis on landscape ruggedness, we performed additional analyses on the subgraphs identified in Figure 1. We applied Fourier decomposition and set all higher-order coefficients to 0. Because removing higher-order coefficients will change the topography of the fitness landscape, such as the number and the location of peaks, we have limited our analysis to the subgraphs in which the quadruple mutant remained the only fitness peak when the higher-order coefficients were removed. Among these subgraphs, we found that the ruggedness score (as determined by the fraction of sign and reciprocal sign epistasis) does not show a systematic trend after removal of higher-order epistasis, and that the number of accessible direct paths can either increase or decrease in the presence of higher-order epistasis (Figure 3—figure supplement 6).

Although it is known that higher-order epistasis increases the variance in fitness, the relation between the degree of higher-order epistasis and the number of accessible direct paths is an open question. In our data, we have shown that the number of accessible direct paths depends on the ruggedness induced by pairwise epistasis (sign and reciprocal sign epistasis, Figure 1—figure supplement 6), which can either go up or down due to the influence of higher-order epistasis. Moreover, we need to consider that higher-order epistasis can mediate indirect paths to increase evolutionary accessibility (Figure 4).

Overall, we conclude that the relation between higher-order epistasis and evolutionary accessibility is an interesting question that warrants future studies, both in theoretical models and in empirical fitness landscapes. We have incorporated our additionalanalyses as figure supplements and expanded our discussion on the role of high-order epistasis in evolutionary accessibility.

“In this study, we observed the presence of higher-order epistasis and systematically quantified its contribution to protein fitness. Our results suggest that higher-order epistasis can either increase or decrease the ruggedness induced by pairwise epistasis, which in turn determines the accessibility of direct paths in a rugged fitness landscape (Figure 3—figure supplement 6). We also revealed the important role of higher-order epistasis in mediating detour bypass, which could promote evolutionary accessibility via indirect paths.”

3) Your simulations allow transitions from every amino acid to any other amino acid and hence ignore the constraints imposed by the encoding as codons. It does not seem to be a big complication to restrict transitions to those that can be achieved by single nucleotide substitutions. Does this reduction in connectivity reduce the effects of indirect path substantially?

We appreciate the question. In the revised manuscript, we have included an analysis that examines the importance of indirect paths in promoting evolutionary accessibility under the constraints imposed by the standard genetic code. Interestingly, this reduction in connectivity did not alter the qualitative finding that indirect paths promote evolutionary accessibility. The result is shown in Figure 4—figure supplement 3 and is described in the text: “We repeated the analysis in Figure 4 with the consideration of the constraints imposed by the standard genetic code (Figure 4—figure supplement 3). The constraints from the genetic code decreased the number of accessible variants due to the reduction in connectivity. However, this reduction in connectivity did not alter our core finding that indirect paths substantially increase evolutionary accessibility (Figure 4—figure supplement 3).”

The methodology of identifying indirect paths under the constraints imposed by the standard genetic code is added to the Materials and methods section: “Under constraints imposed by the standard genetic code, the connectivity of the directed graph that represented the fitness landscape was restricted according to the matrix shown in Figure 4—figure supplement 3. The genetic distance between two variants was calculated according to the matrix shown in Figure 4—figure supplement 3. If the length of the shortest path is larger than the genetic distance between two variants, it means that adaptation requires indirect paths.”

*4) Missing literature and inaccurate statements. Even if there are only two amino acids (or more generally two alleles) per site, there can still be indirect paths with mutational reversions. The statements in the second paragraph of the Main text are not accurate. The mechanisms of bypass and conversion have been discussed previously (and have not been "discovered" by the authors; Main text, fifth paragraph). Gavrilets (1997) discusses extra-dimensional bypass. Several recent papers have theoretically studied the effect of reversions on evolutionary accessibility in diallelic sequence spaces:*

Julien Berestycki, Eric Brunet, Zhan Shi. http://arxiv.org/abs/1401.6894

Anders Martinsson. http://arxiv.org/abs/1501.0220

Li Li. http://arxiv.org/abs/1502.07642

*The adaptation schemes referred to by the authors as 'Greedy model', 'Correlated Fixation Model' and 'Equal Fixation Model' should be placed into their proper population-genetic context. Key references are:*

*H. Allen Orr. Evolution, 56(7), 2002, pp. 1317-1330*

*H. Allen Orr. J. theor. Biol. (2003) 220, 241-247*

which discuss the performance of the three types of 'adaptive walks' in the uncorrelated random 'mutational landscape' model and show, in particular, that greedy walks are much shorter than 'correlated fixation' walks which in turn are shorter than 'equal fixation' walks. This is expected to be a fairly general pattern that also appears in the results shown in Figure 4.

We have followed the reviewers' suggestions and revised our statements and included the suggested references.

The description of reversion-mediated indirect paths in diallelic sequence spaces is added to the revised manuscript: “Most studies of adaptive walks in these diallelic sequence spaces focused on “direct paths” where each mutational step reduces the Hamming distance from the starting point to the destination. However, it has also been shown that mutational reversions can occur during adaptive walks in diallelic sequence spaces such that adaptation proceeds via “indirect paths” (DePristo, Hartl and Weinreich, 2007; Berestycki, Brunet and Shi, 2014; Martinsson, 2015; Li, 2015; Palmer et al., 2015).”

For describing the mechanisms of bypass and conversion, the phrase “we discovered” is replaced with “we observed”, and the appropriate references are included in the revised manuscript: “In sequence space with higher dimensionality (20*^L^*, for a protein sequence with *L* amino acid residues), the extra dimensions may further provide additional routes for adaptation (Gavrilets, 1997; Cariani, 2002). (…) With our experimental data, we observed two distinct mechanisms of bypass, either using an extra amino acid at the same site or using an additional site that allow proteins to continue adaptation when no direct paths were accessible due to reciprocal sign epistasis (Figure 2).”

In the main text of the revised manuscript, the adaptation schemes are described in their population-genetic context: “The Greedy Model represents adaptive evolution of a large population with pervasive clonal interference (de Visser and Krug, 2014). (…) The Equal Fixation Model represents a simplified scenario of adaptation where all beneficial mutations fix with equal probability (Weinreich et al., 2006).”

The suggested references were also included during the discussion of the performance of different types of “adaptive walks”: “Consistent with previous studies (Orr, 2002; Orr, 2003), when mutations conferring larger fitness gains were more likely to fix (e.g. Greedy Model and Correlated Fixation Model), adaptation favored direct moves toward the destination, thus leading to a shorter adaptive paths (Figure 4).”

*5) One take home message of the manuscript is that even in the most rugged place of a protein fitness landscape, the multitude of possible mutations makes most places accessible (via detours at times). This could be a way to reconcile the seemingly contradictory observations that (i) there is a lot of epistasis/ruggedness and (ii) that amino acid preferences are preserved (work by Bloom et al) and that viruses like HIV extensively revert to a putatively optimal fitness peak after immune evasion. In high dimensions, fitness landscapes seem to be locally rugged and accessible at the same time. A more thorough discussion of the potentially wider implications of indirect path could place these results into a broader context.*

Since the fitness landscape described by our work focuses on a small sequence space, we are cautious in extrapolating our results to evolution in a much larger sequence space, such as that of protein homologs. Nonetheless, we agree with the reviewer that the extensive reversion to the putatively optimal fitness peak is consistent with our results here. Therefore, we have extended our Discussion accordingly: “Our work demonstrates that even in the most rugged regions of a protein fitness landscape, most of the sequence space can remain highly accessible owing to the indirect paths opened up by high dimensionality. The enhanced accessibility mediated by indirect paths may provide a partial explanation for some observations in viral evolution. For example, throughout the course of infection HIV always seems to find a way to revert to the global consensus sequence, a putatively “optimal” HIV-1 sequence after immune invasion (Zanini et al., 2015).”